# Simulation-based training using a vessel phantom effectively improved first attempt success and dynamic needle-tip positioning ability for ultrasound-guided radial artery cannulation in real patients: An assessor-blinded randomized controlled study

**Eun Jung Oh[1,2], Jong-Hwan Lee[1], Eun Jin Kwon[1], Jeong Jin Min[1] ***

1 Department of Anesthesiology and Pain Medicine, Samsung Medical Center, Sungkyunkwan University, School of Medicine, Seoul, Korea, 2 Department of Anesthesiology and Pain Medicine, Kangwon National University Hospital, Chuncheon, Korea

* prudence2@hanmail.net

## Abstract

### Background

We evaluated whether a simulation-based training with a vessel phantom improves the basic skills of a novice required for ultrasound-guided radial artery cannulation in real patients. In addition, we analysed whether repeated simulation training sets with an inter-training interval would accelerate the learning curve.

### Methods

From March 2019 to July 2019, twenty-one anesthesiology residents were randomized into either a simulation group (n = 11) or control group (n = 10). Residents performed a total of 84 ultrasound-guided radial artery cannulations in real patients. The simulation group participated in two sets of simulation training on a vessel phantom (10 sessions per set) with a one-month inter-training interval. Trainee's performance proficiency was scored using a developed checklist, and a learning curve for each training set was constructed. To evaluate the effectiveness of our training curriculum in skill transfer, each resident performed four ultrasound-guided radial artery cannulations in real patients. The primary outcome was first attempt success rate and the secondary outcome was dynamic needle-tip positioning ability in real patients.

### Results

The first attempt success rate and dynamic needle-tip positioning ability by ultrasound transducer were significantly higher in the simulation group than the control group (81.8% vs. 50%, $P = 0.002$ and 68.2% vs. 7.5%, $P < 0.001$, respectively). A reduced number of

**Data Availability Statement:** All relevant data are within the paper and its Supporting Information files.

**Funding:** The authors received no specific funding for this work.

**Competing interests:** NO authors have competing interests.

sessions was required to reach a plateau score on the learning curve in the repeated training set compared in the first-set (7 (5–8) vs. 3 (2–4), $P$ = 0.003, respectively).

## Conclusions

Simulation-based training using a vessel phantom effectively improved the first attempt success rate for ultrasound-guided radial artery cannulation in real patients and the dynamic needle-tip positioning ability by ultrasound transducer in novice anesthesiology residents. In addition, repeated training curriculum accelerated the learning curve for recall skill proficiency and reduced inter-individual variability for skill acquisition.

## Clinical trial registration

Clinical Research Information Service (KCT0003471, Principle investigator: Jeong Jin Min, Date of registration: 06/March/2019).

## Introduction

In modern clinical practice, the use of ultrasound-guided technique is rapidly increasing for many procedures. Ultrasound-guided radial artery cannulation has grown in popularity and showed improved success rate at first attempt compared with the palpation method.[1–8] However, ultrasound-guided procedures require the ability to handle an ultrasound machine, proper identification of procedure-relevant anatomy on the ultrasound image, and a combination of visuospatial skills with hand-eye coordination.[9, 10] Therefore, an optimal training curriculum would be helpful to improve the success rate of ultrasound-guided radial artery cannulation.

In several previous studies, simulation-based ultrasound training has improved a trainee's skill proficiency and the skill transferred well to the real-world clinical practice.[11–14] However, to our knowledge, no study has proven the effectiveness of phantom-based simulation training to improve ultrasound-guided radial artery cannulation performance with dynamic needle-tip positioning technique. In this randomized controlled trial, we evaluated whether simulation-based training with a vessel phantom model would improve a novice's basic skills of ultrasound-guided radial artery cannulation in real patients. In addition, we analysed whether repeated simulation training sets with an inter-training interval would accelerate the learning curve.

## Methods

This study was approved by the Samsung Medical Center's Institutional Review Board (SMC 2018-09-085-006, Chairperson Professor Lee Suk-Koo) and written informed consent was obtained from all subjects participating in the trial. The trial was registered prior to patient enrollment at Clinical Research Information Service (KCT0003471, Principal investigator: Jeong Jin Min, Date of registration: 06/March/2019).

### Study population and randomization

From March 2019 to July 2019, anesthesiology residents (1–3 training years) with no experience in ultrasound-guided radial artery cannulation or simulation-based ultrasound-guided phantom training were enrolled in the study. Residents were randomly assigned to either a

**Table 1. Baseline characteristics of anesthesiology residents.**

|  | Simulation group (N = 11) | Control group (N = 10) | P |
|---|---|---|---|
| **Training year (year, 1 / 2 / 3)** | 4 / 4 / 3 | 4 / 3 / 3 | |
| **Palpated artery cannulation (adult, times)** | | | |
| < 50 / 50–100 / ≥ 100 | 1 / 2 / 8 | 1 / 3 / 6 | 0.611 |
| **Palpated artery cannulation (pediatric, times)** | | | |
| None / < 5 / 5–10 / ≥ 10 | 4 / 3 / 1 / 3 | 4 / 2 / 2 / 2 | 0.912 |
| **US-guided central line cannulation (adult, times)** | | | |
| None / 1–10 / 10–50 / > 50 | 0 / 3 / 3 / 5 | 1 / 2 / 3 / 4 | 0.765 |

Values are presented as numbers.

simulation group (n = 11) or a control group (n = 10) using sealed opaque envelopes after stratification based on training year as an anesthesiology resident (Table 1).

To evaluate the effectiveness of simulation training in real clinical practice, adult patients who underwent elective surgeries requiring arterial blood pressure monitoring were enrolled. Patients with a wound near the insertion site, abnormal vascular circulation of the hand (a satisfactory modified Allen test result was ascertained), signs of skin infection, a history of radial artery cannulation within 1 month, and a history of peripheral artery disease were excluded. Independent investigator (E. J. Oh) enrolled the residents and patients for the study and assigned to intervention.

## Equipment

A Blue phantom paediatric four-vessel ultrasound training block model (CAE healthcare®, Sarasota, FL, USA) was used for simulation training. This phantom model was developed for clinician training in psychomotor skills associated with ultrasound vessel cannulation. The model contains four branching blood vessels of various sizes ranging from two to six millimeter. Simulation training was performed on the two millimeter artificial vessel. During the study, all ultrasound procedures were performed with a 13- to 6-MHz linear transducer (Sonosite® M-turbo L25x transducer, Sonosite® Inc., Bothell, WA, USA). Sterile ultrasound transducer covers and sterile ultrasound gels were also used for ultrasound procedures in real patients.

## Study protocol

The study flow in both groups is presented in Fig 1.

## Step 1. Pre-lecture for background knowledge in all residents

Background knowledge regarding ultrasound machine manipulation and ultrasound-guided vascular approach was standardised by a lecture given to residents. The session consisted of one video clip and an oral lecture. The video clip covered knowledge of the ultrasound machine and basic procedural skills of ultrasound-guided radial artery cannulation.[15] The oral lecture was about basic anatomy, preparation of the radial artery insertion site, procedural skills such as out-of-plane needling and transducer manipulation, and research regarding the clinical usefulness of ultrasound-guided radial artery cannulation. After the lecture, all participating residents took a mini quiz to verify that they understood the basic knowledge of the

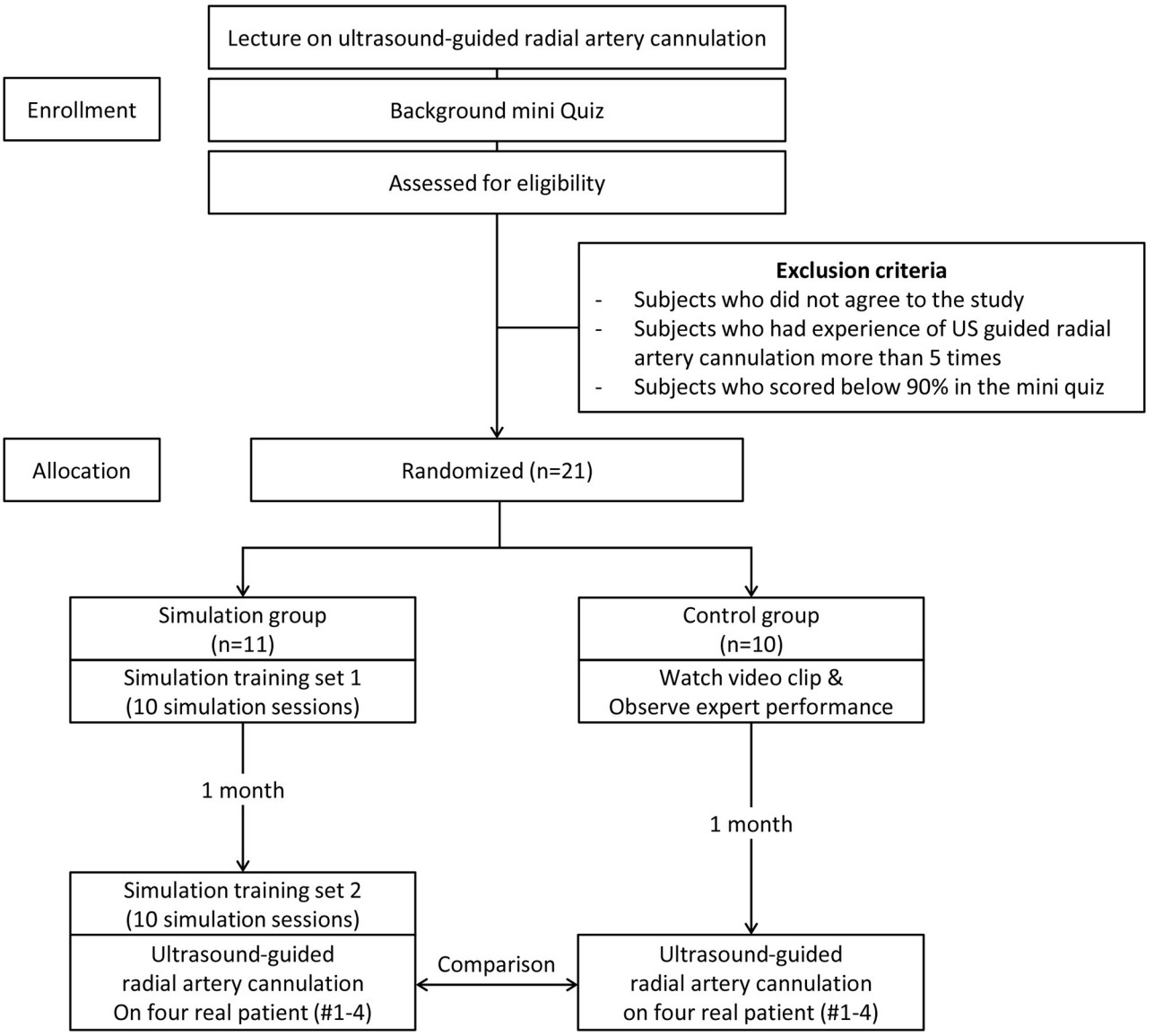

**Fig 1. Flow diagram of study drawn in CONSORT.**

lecture (S1 Table). Residents who scored more than 9 of 10 in the mini quiz proceeded to the next step. All participants scored more than 9 of 10 and were enrolled to the study.

## Step 2. Simulation-based training in simulation group vs. Standard curriculum in control group

The simulation group participated in two simulation training sets with a one-month interval between the sets. Each set consisted of ten simulation training sessions for ability to operate an ultrasound machine, manipulate an ultrasound transducer, and dynamic needle-tip positioning technique on a two millimeter vessel branch in the Blue phantom ultrasound training block (Fig 2). The dynamic needle-tip positioning technique is an ultrasound-guided cannulation technique with an out-of-plane approach. The angiocatheter needle is advanced while

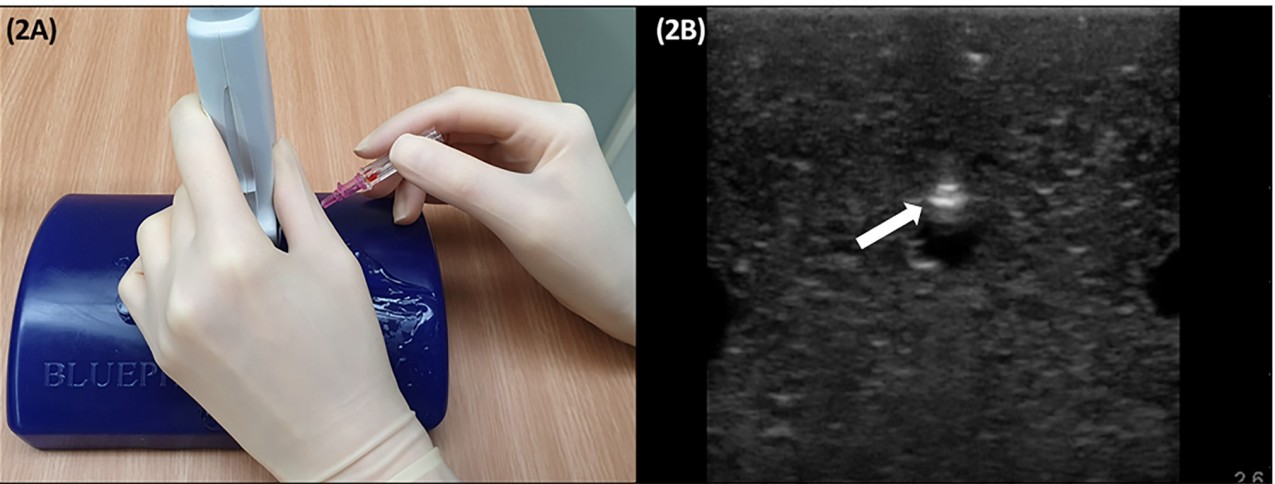

**Fig 2.** (A) Simulation training on Blue phantom paediatric 4 vessel ultrasound training block model. (B) Ultrasound image of the needle tip (hyperechoic dot, white arrow), located at midline of artificial vessel anterior wall. Out-of-plane method (short axis method).

continuously tracing the needle-tip on the ultrasound monitor. Once the hyperechoic dot is visualized between the skin and the artery, the ultrasound transducer is slid slightly in the proximal direction until the hyperechoic dot disappears from the ultrasound monitor (tip position). Subsequently, the needle is advanced until the hyperechoic dot reappears on the ultrasound monitor. This process is repeated until the needle-tip is located on and punctures the anterior wall of the artery and a blood flashback is confirmed on the catheter hub. [16]

Learning objectives were pre-defined as 1) Skill acquirement required in ultrasound machine manipulation for optimal cannulation condition and 2) Skill acquirement to trace the needle-tip in continuous motion with the ultrasound transducer during the procedure (dynamic needle-tip positioning technique) as a major performance milestone. Each simulation session was performed as individual training (1 tutor to 1 participant) for 5 days. The simulation tutor (E. J. Oh) is a skilled anesthesiologist with more than 200 experiences in ultrasound-guided radial artery cannulation using the dynamic needle tip-positioning. In particular, more than half of the experiences of E. J. Oh in ultrasound-guided radial artery cannulation using the dynamic needle tip-positioning were done in children under two years old, who are expected to have procedural difficulties. Also, each year, the tutor has been teaching dynamic needle tip-positioning skills with verbal assistance and skill demonstration as part of the standard curriculum in our tertiary medical center. The tutor gave procedural feedback at the end of each simulation and was not involved in the assessment. All participant simulation procedures were video recorded. After completing two simulation training sets, residents in the simulation group were surveyed on changes in self-confidence for performing the procedure according to a 5-point scale (S2 Table).

An independent investigator (J. J. Min) who was blinded to participant group allocation, reviewed the recorded video clips and assessed participant performance level scores using a pre-developed checklist (S3 Table). This blinded investigator has more than 500 experiences in ultrasound-guided radial arterial cannulation using the dynamic needle tip positioning technique and worked as a tutor in our medical center on practical skills and aspects of performing ultrasound-guided vascular cannulation. Our checklist was developed based on the American Society of Echocardiography and the Society of Cardiovascular Anaesthesiologists recommended training objectives for ultrasound-guided vascular cannulation.[10] The checklist

included ultrasound skills, procedure proficiency, and whether a participant could dynamically position the needle-tip (hyperechoic dot) on the ultrasound image until puncturing the anterior wall of the vessel branch inside the block. Each question was scored as either 1 (performed correctly) or 0 (performed incorrectly) for 17 total questions.

The participants in the control group watched a video clip including operating an ultrasound machine, ultrasound transducer manipulation, and dynamic needle-tip positioning technique The video clip was provided to all participants in the control group, allowing participants to watch whenever they wanted over a month. In addition, during this period the participants in the control group also observed the actual ultrasound-guided radial artery cannulation using the dynamic needle tip-positioning technique by a skilled researcher (E. J. Oh) in more than 10 real patients. This is a standard curriculum for training in ultrasound-guided radial artery cannulation in our medical center.

## Step 3. Evaluation of effectiveness of simulation-based training for skill transfer into clinical practice

One month after study enrolment (when the second set was completed in the simulation group), all residents performed four ultrasound-guided radial artery cannulations in adult patients each. Before the procedure, the radial artery image was recorded to measure artery size, subcutaneous depth, and any anomalies. All radial artery cannulations were performed using an out-of-plane method, as residents were trained. The cannulation performance of each participant and the ultrasound monitor images were video recorded on one screen. After each performance, a blind assessor (J. J. Min) independently reviewed the video clip and scored the performance level score using the same checklist as during the simulation training. Procedure time measurement started when the ultrasound transducer contacted the skin and ended when the arterial waveform was confirmed on the monitor. Procedure time was limited to five minutes on the study protocol. If a procedure took more than five minutes, it was considered as a failure.

The primary outcome was to compare the first attempt success rate of ultrasound-guided radial artery cannulation in real patients between the two groups. It was considered as a cannulation attempt whenever the skin was newly punctured or when a blood flashback on the angiocatheter hub was confirmed. If blood flashback was not seen, the re-directing process of the needle inside the subcutaneous space was defined as a single cannulation attempt. Secondary outcomes included performance level score defined by the checklist score, a participant's dynamic needle-tip tracing ability, procedure time, and total number of attempts. The dynamic needle-tip tracing ability was evaluated based on whether the participant advanced the angiocatheter needle while continuously tracing the needle tip on the ultrasound monitor (successful acquisition of the dynamic needle-tip tracing ability) or identified the hyperechoic dot once during the whole performance (visualized at least once without continuous tracing).

## Statistical analysis

The sample size of real patients was predetermined according to the difference in first attempt success rate between groups. Because there was no previous study on the effect of simulation training in ultrasound-guided radial artery cannulation, sample size was calculated based on the previously reported first attempt success rate in ultrasound-guided radial artery cannulation. The first attempt success rate was 65% in a previous study,[17] while a pilot study in our center involving only novices in ultrasound-guided radial artery cannulation had a first attempt success rate of 35%. Assuming a difference in means of 30% in first attempt success rate between the simulation group and the control group, the necessary minimum sample size

to achieve the desired power of 0.8 and alpha error of 0.05 was 80 artery cannulations (40 per group). To account for a 5% drop out rate, four additional cannulations were recruited. Therefore, the first attempt success rate was measured in a total of eighty-four artery cannulations. Since this study consisted of 21 anesthesiology residents in our tertiary academic medical center, four real patients per resident were recruited. All ultrasound-guided radial artery cannulation on real patients were considered as separate cases.

Statistical analysis was executed using SAS version 9.4 (SAS Institute, Cary, NC) and R 3.4.4 (Vienna, Austria; http://www.R-project.org/). The chi-square test was used to compare the first attempt success rate and the dynamic needle-tip position ability between the two groups. Wilcoxon-signed rank sum test was used for continuous variables measured as secondary outcomes (procedure level score, procedure time, and total attempts). In addition, the learning curve of each simulation set was derived based on procedure level scores using a power model formula of $Y = a \cdot X^b$, where Y represents procedure level score at number of attempt; X represents number of attempt; a represents the time required to produce the first unit of output; and b represents the slope of the learning curve (rate of improvement) when plotted on logarithmic scale.[18, 19] The slopes of the two learning curves were compared by testing the slope of the learning curve with respect to the differences between the two simulation data. $P < 0.05$ was considered significant.

## Results

Twenty-one anesthesiology residents (11 residents in simulation group and 10 residents in control group) participated in and completed the study. Training year, baseline clinical experience of palpated radial artery cannulation, and ultrasound-guided central line cannulation were all comparable between the two groups (Table 1).

### Simulation-based training in simulation group

Learning curves in the simulation group are presented in Fig 3. Compared to the learning curve of the first training set, the learning curve of the repeated training set started with higher performance level score and required fewer training sessions until reaching a plateau score. Therefore, the slopes of the two learning curves differed significantly (0.19 for first training set vs. 0.03 for repeated training set, $P < 0.001$), and the repeated training set showed a narrower confidence interval range reflecting reduced inter-individual variability (1.04 for first training set vs. 0.35 for repeated set) (Fig 3). However, all residents eventually reached full marks based on the performance level checklist within 10 simulation sessions in each set. The median number of sessions required to dynamically position the needle-tip was 4 (3–6) sessions in the first training set and significantly decreased to 1 (1–1) sessions in the second training set ($P = 0.005$). The average number of sessions required to reach the plateau in participant performance level score was 7 (5–8) in the first training set and was reduced to 3 (2–4) sessions in the second training set ($P = 0.003$).

After each simulation training set, all participants self-reported their changes in confidence for ultrasound-guided vascular cannulation performance. Ten of eleven participants reported that they sufficiently improved in self-confidence after the first simulation training set (S2 Table). After completing two sets of simulation training, all 11 participants in the simulation group responded that they were confident enough about the procedure.

### Evaluation of simulation-based skill transfer into clinical practice

Each resident performed four ultrasound-guided radial artery cannulations on real patients. The age of allocated real patients in the control group were significantly younger than the

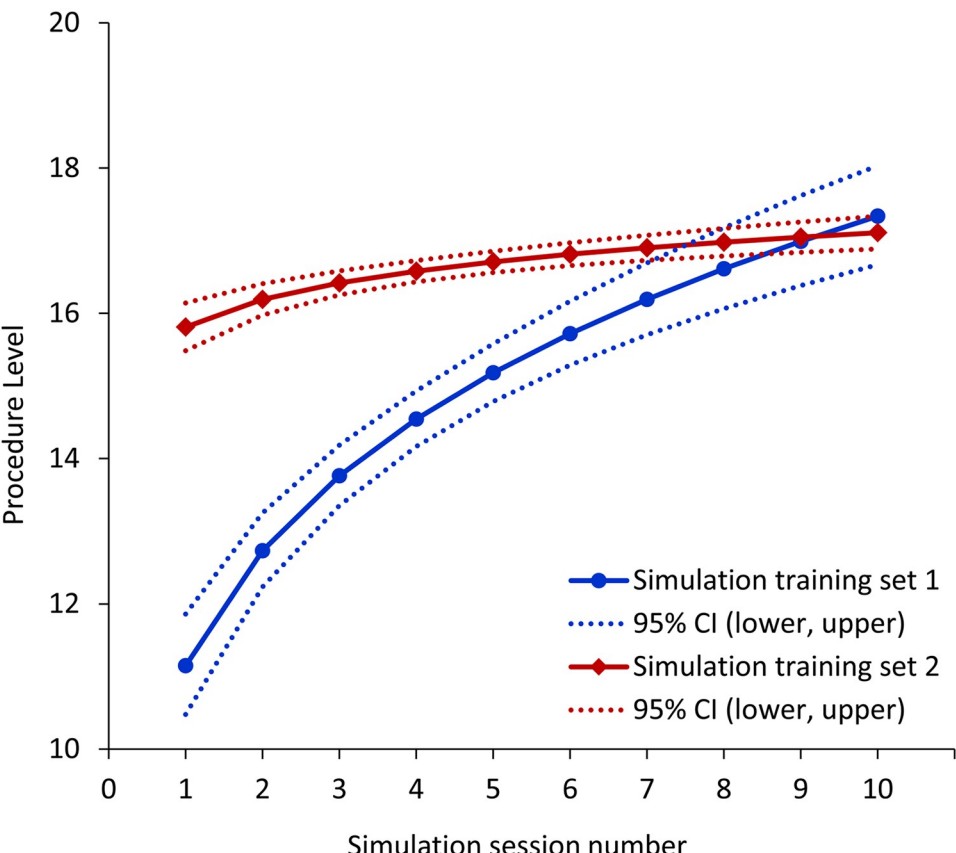

**Fig 3. Learning curves for first simulation training set (blue line) and second simulation training set (red line) after one-month inter-training interval.** CI, confidence interval.

simulation group (52.9 ± 14.4 vs. 61.0 ± 15.2, P = 0.015, respectively). However, the gender distribution and the radial artery characteristics in real patients were comparable between the groups (Table 2). In addition, radial artery diameters in patients were similar to those of vascular branches inside the Blue phantom model used during the simulation training.[20]

**Table 2. Patients characteristics and their radial arterial characteristics.**

|  | Simulation group (N = 44) | Control group (N = 40) | *P* |
|---|---|---|---|
| **Age (years)** | 61.0 ± 15.2 | 52.9 ± 14.4 | 0.015 |
| **Male sex** | 22/44 (50.0) | 19/40 (47.5) | 0.819 |
| **Height (kg)** | 160.9 ± 8.8 | 162.1 ± 10.1 | 0.601 |
| **Weight (cm)** | 65.8 ± 12.5 | 62.9 ± 12.9 | 0.291 |
| **BMI (kg/m$^2$)** | 25.4 ± 4.4 | 23.8 ± 3.6 | 0.074 |
| **Systolic blood pressure (mmHg)** | 125 (106, 142) | 120 (107, 132) | 0.516 |
| **Pulse pressure (mmHg)** | 39 (32, 57) | 46 (32, 62) | 0.446 |
| **Radial artery subcutaneous depth (cm)** | 0.34 ± 0.13 | 0.38 ± 0.16 | 0.168 |
| **Radial artery cross sectional area (cm$^2$)** | 0.05 ± 0.02 | 0.05 ± 0.03 | 0.103 |

Values are presented as mean ± S.D. or median (IQR). Male sex is presented as numbers (proportions). BMI = Body mass index.

**Table 3. Ultrasound-guided radial arterial cannulation performance data in real patients.**

|  | Simulation group (N = 44) | Control group (N = 40) | P |
|---|---|---|---|
| **First attempt success rate** | 36/44 (81.8) | 20/40 (50.0) | 0.002 |
| **Performance level score** | 15.8 ± 1.4 | 13.6 ± 1.7 | < 0.0001 |
| **Successful first attempt cases** | 16.1 ± 1.2 | 14.4 ± 1.4 | <0.0001 |
| **Number of attempts to success 1 / 2 / 3** | 36(81.8) / 7(15.9) / 1(2.3) | 20(54.1) / 16(43.2) / 1(2.7) | 0.017 |
| **Hyperechoic dot (Tip of the needle)** | | | |
| **Visualize at least once** | 44/44 (100.0) | 28/40 (70.0) | <0.0001 |
| **Dynamically positioning ability** | 30/44 (68.2) | 3/40 (7.5) | < 0.0001 |
| **Procedure time, sec** | 65.5 (50.5–150.0) | 134.5 (53.0–226.0) | 0.082 |

Values are presented as mean ± S.D. or median (IQR). First attempt success rate, number of attempts to success, white dot (dynamically positioning ability) are presented as numbers (proportions). Three cases in control group, which failed to successfully place angiocatheter within five minutes, were excluded from the number of attempts to success analysis.

The detailed procedural data in real patients are presented in Table 3. The first attempt success rate in real patients was significantly higher in the simulation group compared to the control group (81.8% vs. 50.0%, $P = 0.002$). Among successful first attempt cases, the average procedure level score was also higher in the simulation group compared to the control group (16.1 ± 1.2 vs. 14.4 ± 1.4, $P < 0.0001$). Even for successful first attempt cases, participants of the control group failed to score points on questions such as whether the participant punctured the midline of the radial artery anterior wall or slid the ultrasound transducer in continuous motion while confirming the ultrasound image of the needle-tip (Dynamic needle-tip positioning ability). In six successful first attempt cases of the control group, the study participant was unable to visualise the ultrasound image of the needle-tip even once (S4 Table). The number of attempts performed until success was also higher in the control group than in the simulation group (Table 3, $P = 0.017$).

During the cannulation procedures, all residents in the simulation group visualised the ultrasound image of the needle-tip at least once, while only 70.0% of residents in the control group identified the ultrasound image of the needle-tip at least once ($P < 0.0001$). The proportion of residents who were able to dynamically position the needle-tip during cannulation was significantly higher in the simulation group compared to the control group (68.2% vs. 7.5%, $P < 0.0001$, respectively). However, there was no significant difference in procedure time between the two groups ($P = 0.082$) (Table 3). Also there were no artery cannulation related complications during the study.

## Discussion

In this randomized controlled trial, we compared the simulation and the control group to demonstrate that training through practice in controlled situations, such as simulation, is more effective in transferring skills to actual performance than training through observation. Two sets of simulation-based training significantly improved the first attempt success rate and the dynamic needle-tip positioning ability on ultrasound-guided radial artery cannulation in real patients compared to the control group residents who were trained through watching a video clip on ultrasound-guided radial artery cannulation and observing actual performances of a skilled researcher. In addition, repeated training curriculum accelerated the learning curve for recall skill proficiency and reduced inter-individual variability for skill acquisition among residents in the simulation group.

The simulation-based training curriculum of the present study was constructed to include ten simulation sessions in each training set. All residents in the simulation group reached the plateau in performance level score before ten simulation sessions were completed. Ten simulation sessions were sufficient to demonstrate improvement in skill proficiency, which is in agreement with the findings of previous study suggesting range of six to ten simulation training sessions to demonstrate competence in ultrasound-guided radial artery cannulation.[10]

In addition, our curriculum was designed with repeated training sets and an one month inter-training interval based on previous results that skill acquisition is influenced by training distribution.[21] Residents retained the skill better when they were taught in a repeated training manner with a period of rest between sessions compared to a mass training including all training at a single session.[22, 23] This is thought to be a consequence of the motion neural process that continues during the rest period between training sessions.[24] In the present study, psychomotor skills were retained after the inter-training interval between simulation sets. Skill proficiency in ultrasound-guided cannulation of the vascular model improved as simulation sessions progressed in both training sets. However, a resident's procedure level score reached the plateau after fewer training sessions and inter-individual variability decreased in the second simulation set. This result supports the importance of repeating simulation training with interval.

The effectiveness of simulation training curriculum to transfer skill into clinical practice was assessed through ultrasound-guided radial artery cannulation performance in real adult patients. Residents in the simulation group showed a similar first attempt success rate as the previously reported success rates of 71.4% in experienced cardiac anesthesiologists [25] and 83% in faculty anesthesiologists. On the other hand, the residents in the control group showed a similar first attempt success rates of 53% in anesthesia trainees with less than five experiences in ultrasound-guided radial artery cannulation [26] and 62% in anesthetists with ultrasound-guided central vein insertion experience but novice to ultrasound-guided radial artery cannulation.[27] Among successful first attempt cases, the control group showed less success in terms of dynamic needle-tip positioning ability during the procedure, and this group achieved lower overall procedure level scores than the simulation group. It can be deduced that some of the successful cases in the control group were carried out without following the key elements of ultrasound-guided procedures emphasized in this study. The key elements, such as dynamically positioning the needle-tip or midline puncture of the radial artery anterior wall are known to increase the success rate.[28] Thus, ultrasound-guided procedures which were performed with missing the key elements may be difficult to evaluate as proficient skill acquisition of ultrasound-guided cannulation.

Unexpectedly, procedure time was not significantly different between groups. Participants in the simulation group tended to follow knowledge acquired from the simulation training, such as moving the transducer and the angiocatheter in continuous motion guided by the hyperechoic image of the needle-tip on the ultrasound monitor. Consequently, the procedure was performed more accurately, but the procedure time was longer. Moreover, procedure time varied depending on each participant's character. Some participants were more hesitant to advance the needle even while looking at the needle-tip in real time. These individual differences make procedure time an inaccurate tool to assess proficiency.[21]

Although several studies have investigated simulation-based training for ultrasound-guided procedures, there was no single optimised training curriculum for ultrasound-guided radial artery cannulation. To the best of our knowledge, the present study is the first randomized controlled trial to verify the Blue phantom vascular block model as a simulator for ultrasound-guided radial artery cannulation and to evaluate the different forms of training on ultrasound-guided radial artery cannulation with dynamic needle-tip positioning. The strength of this

study is that performance assessments were conducted in an actual workplace among real patients by recorded video clips. Based on 'Miller's pyramid of competence', the workplace-based assessment results show a significant correlation with skill proficiency in clinical practice. [29] Thus, this study may provide a basis for constructing a standard simulation training curriculum for ultrasound-guided radial artery cannulation which leads to enhanced clinical performance.

This study has several limitations. First, it is limited to a single center with a small number of participating residents. The number of residents in our department is an unchangeable condition that we tried to overcome by having each participant perform four ultrasound-guided radial artery cannulations in real patients resulting in eighty-four cannulations in the study. Second, a non-validated checklist was used to score participant performance level because there are no standardised assessment tools for ultrasound-guided radial artery cannulation performance. However, checklist questionnaires were objectively constructed based on the American Society of Echocardiography and the Society of Cardiovascular Anaesthesiologists recommended training objectives for ultrasound-guided vascular cannulation. In addition, the same score was given for all questions on the checklist; however, some questions regarding the main learning points may be weighted in future studies. Fourth, the age of real patients allocated in the simulation group were significantly older compared to the control group. This age difference between the two groups may be due to random chance in a finite sample. However, factors that can be associated with catheterization failures, such as artery tortuosity, or presence of atherosclerosis, usually increase with age. [30, 31] Thus, we suspect that the influence of age even supports the effectiveness of simulation training which is the conclusion of our study. Finally, we did not observe the long-term effect of simulation-based training. The participating residents continue clinical practice every day, which may improve their procedure skills and making it difficult to evaluate the sole effect of the simulation training curriculum.

## Conclusion

Simulation-based training using a Blue phantom vascular block model effectively improved the first attempt success rate for ultrasound-guided radial artery cannulation as well as dynamic needle-tip positioning technique with an ultrasound transducer scan in novice anesthesiology residents. In addition, repeated training curriculum accelerated the learning curve for recall skill proficiency and reduced inter-individual variability for skill acquisition.

## Supporting information

**S1 Table. Quiz for background knowledge on ultrasound-guided radial artery cannulation.** (DOCX)

**S2 Table. 5-point scale for assessing the change in self confidence after the simulation training set.** (DOCX)

**S3 Table. Checklist for procedures of ultrasound-guided radial artery cannulation.** (DOCX)

**S4 Table. The questions that failed to score points on the checklist among successful first attempt cases.** (DOCX)

**S1 Data.** (XLSX)

## Author Contributions

**Conceptualization:** Jeong Jin Min.

**Data curation:** Eun Jung Oh, Eun Jin Kwon, Jeong Jin Min.

**Formal analysis:** Eun Jung Oh, Jong-Hwan Lee, Eun Jin Kwon, Jeong Jin Min.

**Methodology:** Eun Jung Oh, Jeong Jin Min.

**Project administration:** Eun Jung Oh, Jeong Jin Min.

**Supervision:** Jong-Hwan Lee.

**Writing – original draft:** Eun Jung Oh, Jong-Hwan Lee, Jeong Jin Min.

**Writing – review & editing:** Eun Jung Oh, Jong-Hwan Lee, Jeong Jin Min.

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
