## [Decision Letter · Decision Letter 0]

23 Mar 2020

PONE-D-19-35381

Simulation-based training using a vessel phantom effectively improved first attempt success and dynamic needle-tip positioning ability for ultrasound-guided radial artery cannulation in real patients: an assessor-blinded randomized controlled study

PLOS ONE

Dear Dr. Min,

Thank you for submitting your manuscript to PLOS ONE. After careful consideration, we feel that it has merit but does not fully meet PLOS ONE’s publication criteria as it currently stands. Therefore, we invite you to submit a revised version of the manuscript that addresses the points raised during the review process.

The manuscript has been assessed by two reviewers; their comments are available below.

The reviewers have raised some major items that need attention in a revision, the reviewers feel that further information should be provided on the training that each group received, and they have also requested additional clarification on the secondary outcome measured and a number of aspects of the methodology.

Could you please carefully revise the manuscript to address the comments raised by the reviewers?

We would appreciate receiving your revised manuscript by May 05 2020 11:59PM. Please include the following items when submitting your revised manuscript:

We look forward to receiving your revised manuscript.

Kind regards,

Iratxe Puebla

Deputy Editor-in-Chief, PLOS ONE

Journal Requirements:

Reviewers' comments:

Reviewer's Responses to Questions

**Comments to the Author**

1. Is the manuscript technically sound, and do the data support the conclusions?

Reviewer #1: Yes

Reviewer #2: Yes

2. Has the statistical analysis been performed appropriately and rigorously? 

Reviewer #1: Yes

Reviewer #2: I Don't Know

3. Have the authors made all data underlying the findings in their manuscript fully available?

Reviewer #1: Yes

Reviewer #2: Yes

4. Is the manuscript presented in an intelligible fashion and written in standard English?

Reviewer #1: Yes

Reviewer #2: Yes

5. Review Comments to the Author

Reviewer #1: Thank you for allowing me to review " Simulation-based training using a vessel phantom effectively improved first attempt success and dynamic needle-tip positioning ability for ultrasound-guided radial artery cannulation in real patients: an assessor-blinded randomized controlled study" by Oh et al.

The authors compared first success rate of ultrasound guided artery cannulation by resident between simulation training versus control groups. The first attempt success rate and dynamic needle-tip positioning ability by ultrasound transducer were significantly higher in the simulation group than the control group.

This is very interesting study.

My suggestions are as follows.

The authors should describe the detail of dynamic needle tip positioning in the method section referring to previous reports.

Dynamic needle-tip tracing ability as the secondary outcome is unclear. Please clarlify.

There are terms of “procedure duration time”, “procedure duration”, and “procedure time”. Please unify the expression as it is confusing.

The question 1 and 2 of S1 table seem like the same question.

Table 3

Performance level→Performance lever score in successful first attempt cases?

Why two data are described? 16 (14.5-17) and 16.1 ± 1.2

Procedure time→please unify

Figure 2B

Please point the needle tip by arrow or something.

Reviewer #2: The paper does not make it very clear what types of training are being compared? The performance of the dynamic needle tip positioning technique was evaluated but more detailed information is needed on the actual training received by the residents.

Line 126: It is not clear what the experience level of the proctors was as far as the dynamic needle tip positioning technique.

Line 132: Did the video clips that were used for grading the participants show the resident technique or the ultrasound images or both. You would need a recording of the ultrasound images to properly grade the dynamic needle tip technique

Line 138: What is the experience level of the researcher and what ultrasound cannulation technique was he/she teaching the residents

Line 152: Was the timing of the procedure started upon contact of the ultrasound probe and then skin or upon puncturing of the skin by the needle?

Line 190: There was a statistically significant difference in the median ages of the patients in the simulation and the control group. This should be mentioned in the results section and in the discussion

Line 190: Gender has been shown to be an important factor affecting success of radial artery cannulation. This should have been looked at in the baseline characteristics table

Line 286: To draw a conclusion that the control group achieved radial arterial cannulation by chance is incorrect. They simply used a different ultrasound technique

6. PLOS authors have the option to publish the peer review history of their article (what does this mean?). If published, this will include your full peer review and any attached files.

Reviewer #1: No

Reviewer #2: No

---

## [Author Response · Author response to Decision Letter 0]

20 Apr 2020

April 07, 2020

Simulation-based training using a vessel phantom effectively improved first attempt success and dynamic needle-tip positioning ability for ultrasound-guided radial artery cannulation in real patients: an assessor-blinded randomized controlled study

Dear Editor

We thank you and the reviewers of Plos One for taking the time to review our article. We made corrections and clarifications to the manuscript to address the reviewer comments. All authors have read and approved the revised manuscript. The changes are summarized below.

Point-to-Point Responses to The First Reviewer’s comments

Reviewer #1

Thank you for allowing me to review " Simulation-based training using a vessel phantom effectively improved first attempt success and dynamic needle-tip positioning ability for ultrasound-guided radial artery cannulation in real patients: an assessor-blinded randomized controlled study" by Oh et al.

The authors compared first success rate of ultrasound guided artery cannulation by resident between simulation training versus control groups. The first attempt success rate and dynamic needle-tip positioning ability by ultrasound transducer were significantly higher in the simulation group than the control group. This is very interesting study.

1. The authors should describe the detail of dynamic needle tip positioning in the method section referring to previous reports. Dynamic needle-tip tracing ability as the secondary outcome is unclear. Please clarlify.

Response: We have added the dynamic needle-tip positioning method and how we evaluated dynamic needle-tip tracing ability as a secondary outcome in the revised Methods section as follows.

“The dynamic needle-tip positioning technique is an ultrasound-guided cannulation technique with an out-of-plane approach. The angiocatheter needle is advanced while continuously tracing the needle-tip on the ultrasound monitor. Once the hyperechoic dot is visualized between the skin and the artery, the ultrasound transducer is slid slightly in the proximal direction until the hyperechoic dot disappears from the ultrasound monitor (tip position). Subsequently, the needle is advanced until the hyperechoic dot reappears on the ultrasound monitor. This process is repeated until the needle-tip is located on and punctures the anterior wall of the artery and a blood flashback is confirmed on the catheter hub. (1) ” In lines 122-128. 

“The dynamic needle-tip tracing ability was evaluated based on whether the participant advanced the angiocatheter needle while continuously tracing the needle tip on the ultrasound monitor (successful acquisition of the dynamic needle-tip tracing ability) or identified the hyperechoic dot once during the whole performance (visualized at least once without continuous tracing).” In lines 176-180.

2. There are terms of “procedure duration time”, “procedure duration”, and “procedure time”. Please unify the expression as it is confusing. 

Response: We appreciate your comment. We made the expression consistent as “procedure time” throughout the manuscript. 

3. The question 1 and 2 of S1 table seems like the same question.

Response: Thank you for your comments. We have corrected the second question as follows.

“ 2) The clinician should place the ultrasound machine directly across from the operator to establish the best ergonomic position for the procedure.” (Supplementary Table 1). 

4. In Table 3, Performance level→Performance level score in successful first attempt cases?

Why two data are described? 16 (14.5-17) and 16.1 ± 1.2

Response: After checking the data distribution normality, we have revised the performance level score presentation for the mean ± standard deviation (S.D.) in revised Table 3 as follows:

 Simulation group (N=44) Control group (N=40) P-value

Performance level score

Successful first attempt cases 15.8 ± 1.4

16.1 ± 1.2 13.6 ± 1.7

14.4 ± 1.4 < 0.001

<0.001

5. In Figure 2B, Please point the needle tip by arrow or something.

Response: Thank you for your comments. We have added a new white arrow to Figure 2B to point to the needle tip. 

Reviewer #2

1. The paper does not make it very clear what types of training are being compared? The performance of the dynamic needle tip positioning technique was evaluated but more detailed information is needed on the actual training received by the residents.

Response: We appreciate the reviewer for this critical comment. Currently, in the information era, there are numerous educational materials/videos available online that a trainee could view to obtain knowledge about an invasive procedure before performing it on real patients. In comparison to just knowing how to do it by learning through various educational materials or observing the expert’s practices (control group), we aimed to evaluate the effectiveness of simulation-based learning in which trainees actually practice the procedure several times before performing it on real patients (simulation group). Therefore, we compared the simulation group and the control group to demonstrate that training through practice in controlled situations, such as simulation training, is more effective in transferring skills to actual performance than training through observation. In our study, the simulation group underwent two sets of simulation training of ultrasound-guided radial artery cannulation on a phantom vascular model. On the other hand, the control group learned the technique through education materials/video clips or observing actual clinical performances by a skilled researcher. 

Specifically, the contents of the ultrasound-guided radial artery training were ability to operate an ultrasound machine, ultrasound transducer manipulation without compressing the procedure relevant anatomy, hand-eye coordination, and dynamic needle-tip positioning technique. 

After the two different forms of training in each group, we evaluated the effectiveness of education type by first attempt success rate in ultrasound-guided radial artery cannulation among real patients. Based on ‘Miller’s pyramid for assessing clinical competence’, the performance assessment was carried out at the actual workplace among real patients. We video recorded the performance and the video clip was assessed using a pre-developed checklist. The pre-developed checklist included ultrasound skills, procedure proficiency, and whether a participant could dynamically position the needle-tip (hyperechoic dot) on the ultrasound image until puncturing the anterior wall of the radial artery. We have clarified the training method in the revised Discussion section. 

“In this randomized controlled trial, we compared the simulation and control groups to demonstrate that training through practice in controlled situations, such as simulation, is more effective in transferring skills to actual performance than training through observation. Two sets of simulation-based training significantly improved the first attempt success rate and the dynamic needle-tip positioning ability on ultrasound-guided radial artery cannulation in real patients compared to the control group residents who were trained through watching a video clip on ultrasound-guided radial artery cannulation and observing actual performances of a skilled researcher.” In lines 270-276. 

“To the best of our knowledge, the present study is the first randomized controlled trial to verify the Blue phantom vascular block model as a simulator for ultrasound-guided radial artery cannulation and to evaluate the different forms of training on ultrasound-guided radial artery cannulation with dynamic needle-tip positioning. A strength of this study is that performance assessments were conducted in an actual workplace among real patients by recorded video clips. Based on ‘Miller’s pyramid of competence’, the workplace-based assessment results show a significant correlation with skill proficiency in clinical practice.(2) Thus, this study may provide a basis for constructing a standard simulation training curriculum for ultrasound-guided radial artery cannulation which leads to enhanced clinical performance.” In lines 319-327.

Also, we have added detailed training content to the revised Methods section.

 “Each set consisted of ten simulation training sessions for ability to operate an ultrasound machine, manipulate an ultrasound transducer, and dynamic needle-tip positioning technique on a two millimeter vessel branch in the Blue phantom ultrasound training block” in lines 119-121 and “Participants in the control group watched a video clip including operating an ultrasound machine, ultrasound transducer manipulation, and dynamic needle-tip positioning technique, and they observed the actual performances of a skilled researcher (E. J. Oh) on real patients for a month.” in lines 151-153.

2. It is not clear what the experience level of the proctors was as far as the dynamic needle tip positioning technique.

Response: The blinded proctor (J. J. Min) in our study is an expert in cardiac anesthesiology with more than 500 experiences in ultrasound-guided vascular cannulation using the dynamic needle tip positioning technique. She has conducted a study comparing ultrasound-guided with palpation-guided techniques for radial arterial catheterization in infants (3) and studies related to ultrasound-guided vascular cannulation using the dynamic needle-tip positioning technique. We have added the information on proctor experience level to the revised Methods section. 

“An independent investigator (J. J. Min) who was blinded to participant group allocation, reviewed the recorded video clips and assessed participant performance level scores using a pre-developed checklist (S3 Table). This blinded investigator has more than 500 experiences in ultrasound-guided radial arterial cannulation using the dynamic needle tip positioning technique and worked as a tutor in our medical center on practical skills and aspects of performing ultrasound-guided vascular cannulation.” In lines 139-144.

3. Did the video clips that were used for grading the participants show the resident technique or the ultrasound images or both. You would need a recording of the ultrasound images to properly grade the dynamic needle tip technique.

Response: The participating resident’s procedure technique and the ultrasound monitor image were simultaneously recorded on one screen with a video camera. Therefore, the blinded assessor (J. J. Min) was able to properly score if the residents performed dynamic needle-tip positioning accurately. We revised the indicated sentences in the Methods section. 

“The cannulation performance of each participant was video recorded.” to “The cannulation performance of each participant and the ultrasound monitor images were video recorded on one screen.” In lines 163-165. 

4. What is the experience level of the researcher and what ultrasound cannulation technique was he/she teaching the residents. 

Response: The simulation tutor (E. J. Oh) in our study is a skilled anesthesiologist with more than 200 experiences in ultrasound-guided radial artery cannulation and more than 400 experiences in ultrasound-guided central venous catheter insertion. She has conducted studies related to ultrasound-guided vascular cannulation using the dynamic needle-tip positioning technique and also performed individual training (1 tutor to 1 participant) in a tertiary medical center. The individual training covered practical skills and aspects of performing ultrasound-guided vascular cannulation, i.e., combination of visuospatial skills with hand-eye coordination, dynamic needle-tip positioning technique, and two puncture techniques (anterior/posterior). We have added more details about this to the revised Methods section as follows:

“The dynamic needle-tip positioning technique is an ultrasound-guided cannulation technique with an out-of-plane approach. The angiocatheter needle is advanced while continuously tracing the needle-tip on the ultrasound monitor. Once the hyperechoic dot is visualized between the skin and the artery, the ultrasound transducer is slid slightly in the proximal direction until the hyperechoic dot disappears from the ultrasound monitor (tip position). Subsequently, the needle is advanced until the hyperechoic dot reappears on the ultrasound monitor. This process is repeated until the needle-tip is located on and punctures the anterior wall of the artery and a blood flashback is confirmed on the catheter hub.(1)” in lines 122-128 and “The simulation tutor (E. J. Oh) is a skilled anesthesiologist with more than 200 experiences in ultrasound-guided radial artery cannulation.” In lines 133-134.

5. Was the timing of the procedure started upon contact of the ultrasound probe and then skin or upon puncturing of the skin by the needle?

Response: The “procedure time” measurement started when the ultrasound transducer contacted the skin and ended when the arterial waveform was confirmed on the monitor. We have clarified this in the revised manuscript.

“Procedure time measurement started when the ultrasound transducer contacted the skin and ended when the arterial waveform was confirmed on the monitor.” In lines 167-168, and “It was considered as a cannulation attempt whenever the skin was newly punctured or when a blood flashback on the angiocatheter hub was confirmed. If blood flashback was not seen, the re-directioning process of the needle inside the subcutaneous space was defined as a single cannulation attempt.” In lines 171-174.

6. There was a statistically significant difference in the median ages of the patients in the simulation and the control group. This should be mentioned in the results section and in the discussion section.

Response: As the reviewer suggested we have added information on real patient age to the revised Results section and in the revised Discussion section. 

“The age of allocated real patients in the control group were significantly younger than the simulation group (52.9 ± 14.4 vs. 61.0 ± 15.2, P = 0.015, respectively).” In lines 232-234. 

“Fourth, the age of real patients allocated in the simulation group were significantly older compared to the control group. This age difference between the two groups may be due to random chance in a finite sample. However, factors that can be associated with catheterization failures, such as artery tortuosity, or presence of atherosclerosis, usually increase with age. (4, 5) Thus, we suspect that the influence of age even supports the effectiveness of simulation training which is the conclusion of our study.” In lines 337-342.

7. Gender has been shown to be an important factor affecting success of radial artery cannulation. This should have been looked at in the baseline characteristics table.

Response: Thank you for your comments. We are aware that gender affects the vascular access ability and failure rate for arterial catheter insertion. (6, 7) In our study, the distribution of gender did not differ between the simulation group and the control group (P = 0.819). We have added the data on gender distribution of the real patient participants to the revised Table 2 as follows and in the revised Results section.

Table 2. Patients characteristics and their radial arterial characteristics.

 Simulation group (N=44) Control group (N=40) P

Male sex 22/44 (50.0) 19/40 (47.5) 0.819

“However, the gender distribution and the radial artery characteristics in real patients were comparable between the groups (Table 2).” In lines 234-235.

8. To draw a conclusion that the control group achieved radial arterial cannulation by chance is incorrect. They simply used a different ultrasound technique.

Response: We fully agree with your suggestion. We have changed the description in the revised Discussion section according to your comments.

“However, among successful first attempt cases, the control group showed less success in terms of dynamic needle-tip positioning ability during the procedure, and this group achieved lower overall procedure level scores than the simulation group. It can be deduced that some of the successful cases in the control group were carried out without following the key elements of ultrasound-guided procedures emphasized in this study. The key elements, such as dynamically positioning the needle-tip or midline puncture of the radial artery anterior wall are known to increase the success rate. (8) Thus, ultrasound-guided procedures which were performed with missing the key elements may be difficult to evaluate as proficient skill acquisition of ultrasound-guided cannulation.” In lines 301-308.

References

1. Clemmesen L, Knudsen L, Sloth E, Bendtsen T. Dynamic needle tip positioning - ultrasound guidance for peripheral vascular access. A randomized, controlled and blinded study in phantoms performed by ultrasound novices. Ultraschall in der Medizin (Stuttgart, Germany : 1980). 2012;33(7):E321-e5.

2. Wass V, Van der Vleuten C, Shatzer J, Jones R. Assessment of clinical competence. The Lancet. 2001;357(9260):945-9.

3. Min JJ, Tay CK, Gil NS, Lee JH, Kim S, Kim CS, et al. Ultrasound-guided vs. palpation-guided techniques for radial arterial catheterisation in infants: A randomised controlled trial. European journal of anaesthesiology. 2019;36(3):200-5.

4. Lee D, Kim JY, Kim HS, Lee KC, Lee SJ, Kwak HJ. Ultrasound evaluation of the radial artery for arterial catheterization in healthy anesthetized patients. Journal of clinical monitoring and computing. 2016;30(2):215-9.

5. Crow MT. Atherosclerosis and the vascular biology of aging. Aging Clinical and Experimental Research. 1996;8(4):221-34.

6. Santen SA, Yamazaki K, Holmboe ES, Yarris LM, Hamstra SJ. Comparison of Male and Female Resident Milestone Assessments During Emergency Medicine Residency Training: A National Study. Acad Med. 2020;95(2):263-8.

7. Eisen LA, Minami T, Berger JS, Sekiguchi H, Mayo PH, Narasimhan M. Gender disparity in failure rate for arterial catheter attempts. Journal of intensive care medicine. 2007;22(3):166-72.

8. Takeshita J, Yoshida T, Nakajima Y, Nakayama Y, Nishiyama K, Ito Y, et al. Dynamic Needle Tip Positioning for Ultrasound-Guided Arterial Catheterization in Infants and Small Children With Deep Arteries: A Randomized Controlled Trial. Journal of cardiothoracic and vascular anesthesia. 2019;33(7):1919-25.

We hope the revised manuscript will better meet your requirements for publication. We thank the editor and the reviewers of Plos One once again for their constructive review of our paper. 

Warm regards,

Jeong Jin Min, M.D., Ph.D. 

Samsung Medical Center,

Sungkyunkwan University School of Medicine

---

## [Decision Letter · Decision Letter 1]

19 May 2020

PONE-D-19-35381R1

Simulation-based training using a vessel phantom effectively improved first attempt success and dynamic needle-tip positioning ability for ultrasound-guided radial artery cannulation in real patients: an assessor-blinded randomized controlled study

PLOS ONE

Dear Dr Jeong Jin Min,

Thank you for submitting your manuscript to PLOS ONE. After careful consideration, we feel that it has merit but does not fully meet PLOS ONE’s publication criteria as it currently stands. Therefore, we invite you to submit a revised version of the manuscript that addresses the points raised during the review process.

ACADEMIC EDITOR: 

The authors adressed all of my concern (as reviewer 1). They still should adress the minor comment of reviewer 2. 

We would appreciate receiving your revised manuscript by Jul 03 2020 11:59PM. To enhance the reproducibility of your results, we recommend that if applicable you deposit your laboratory protocols in protocols.io, where a protocol can be assigned its own identifier (DOI) such that it can be cited independently in the future. For instructions see: http://journals.plos.org/plosone/s/submission-guidelines#loc-laboratory-protocols

We look forward to receiving your revised manuscript.

Kind regards,

Jun Takeshita, M.D., Ph.D.

Academic Editor

PLOS ONE

Reviewers' comments:

Reviewer's Responses to Questions

**Comments to the Author**

1. If the authors have adequately addressed your comments raised in a previous round of review and you feel that this manuscript is now acceptable for publication, you may indicate that here to bypass the “Comments to the Author” section, enter your conflict of interest statement in the “Confidential to Editor” section, and submit your "Accept" recommendation.

Reviewer #2: (No Response)

2. Is the manuscript technically sound, and do the data support the conclusions?

Reviewer #2: Yes

3. Has the statistical analysis been performed appropriately and rigorously? 

Reviewer #2: Yes

4. Have the authors made all data underlying the findings in their manuscript fully available?

Reviewer #2: Yes

5. Is the manuscript presented in an intelligible fashion and written in standard English?

Reviewer #2: Yes

6. Review Comments to the Author

Reviewer #2: Dear Editor,

Thank you for the opportunity to review this manuscript. The authors have done a great job or responding to the past reviewer comments. My additional comments are:

1. Line 134-135 The authors responded by clarifying the experiences of E.J. Oh but because this is a study evaluating dynamic needle tip positioning, they should also add what the dynamic needle tip positioning experience of this provider was.

2. Line 153-154 Because this is an evaluation of simulation training vs other training methods, it should be clear what the control group experience was. How many procedures on average did they participants in the control group witness? Were these witnessed experiences all dynamic needle tip positioning or not?

3. Line 165-166 It seems like this line was edited but both the original and edited sentences remained in the manuscript. Please correct this

4. Line 212 "was all comparable" should read "were all comparable"

5. Supplement table 4. It is not clear how to read this table. Because these are all questions, maybe it would be better to create "yes" and "no" columns for the control and the simulation group and then fill in the number of responses

6. Line 284. The word "enhance" does not fit in the sentence well. Consider using enhancement, improvement, etc

7. Line 302. The 50% first pass success rate in novices is not higher than expected. Ueda et al in A randomized controlled trial of radial artery cannulation guided by doppler vs palpation vs ultrasound, achieved a first pass of 53% in anesthesia trainees with < 5 USG radial artery cannulations. levin et al in Use of ultrasound guidance in the insertion of radial artery catheters, Crit Care Med 2003, achieved a first pass success rate of 62% in operators with ultrasound guided central venous catheter experience but novices in radial artery cannulation with ultrasound guidance

7. PLOS authors have the option to publish the peer review history of their article (what does this mean?). If published, this will include your full peer review and any attached files.

Reviewer #2: No

---

## [Author Response · Author response to Decision Letter 1]

26 May 2020

May, 26, 2020

Simulation-based training using a vessel phantom effectively improved first attempt success and dynamic needle-tip positioning ability for ultrasound-guided radial artery cannulation in real patients: an assessor-blinded randomized controlled study

Dear Editor

We thank you and the reviewers of Plos One for taking the time to review our article once again. We made corrections and clarifications to the manuscript to address the reviewer comments. All authors have read and approved the revised manuscript. The changes are summarized below.

Reviewer #2

1. Line 134-135. The authors responded by clarifying the experiences of E.J. Oh but because this is a study evaluating dynamic needle tip positioning, they should also add what the dynamic needle tip positioning experience of this provider was.

Response: As the reviewer suggested we have added information on dynamic needle tip positioning experience of the simulation tutor (E. J. Oh) in our study. The simulation tutor (E. J. Oh) has more than 200 experiences in ultrasound-guided radial artery cannulation using the dynamic needle tip-positioning and have conducted a study on ultrasound-guided vascular cannulation using the dynamic needle-tip positioning technique in children under two year old. Also, each year, the tutor has been teaching dynamic needle tip-positioning skills with verbal assistance and skill demonstration as part of the standard curriculum in our tertiary medical center. We have added more details about this to the revised Methods section as follows:

 “The simulation tutor (E. J. Oh) is a skilled anesthesiologist with more than 200 experiences in ultrasound-guided radial artery cannulation using the dynamic needle tip-positioning. In particular, more than half of experiences of E. J. Oh in ultrasound-guided radial artery cannulation using the dynamic needle tip-positioning were done in children under two years old, who are expected to have procedural difficulties. Also, each year, the tutor has been teaching dynamic needle tip-positioning skills with verbal assistance and skill demonstration as part of the standard curriculum in our tertiary medical center.” In lines 133-139.

2. Line 153-154. Because this is an evaluation of simulation training vs other training methods, it should be clear what the control group experience was. How many procedures on average did they participants in the control group witness? Were these witnessed experiences all dynamic needle tip positioning or not?

Response: We fully agree with your comments. The participants in the control group watched a video clip including operating an ultrasound machine, ultrasound transducer manipulation, and dynamic needle-tip positioning technique. The video clip was provided to all participants in the control group, allowing participants to watch whenever they wanted. In addition, the participants in the control group observed the actual performances of a skilled researcher (E. J. Oh) in more than 10 real patients over a month. We have added the information on proctor experience level to the revised Methods section. 

“The participants in the control group watched a video clip including operating an ultrasound machine, ultrasound transducer manipulation, and dynamic needle-tip positioning technique. The video clip was provided to all participants in the control group, allowing participants to watch whenever they wanted over a month. In addition, during this period the participants in the control group also observed the actual ultrasound-guided radial artery cannulation using the dynamic needle tip-positioning by a skilled researcher (E. J. Oh) in more than 10 real patients.” In lines 156-161.

3. Line 165-166. It seems like this line was edited but both the original and edited sentences remained in the manuscript. Please correct this

Response: We appreciate the reviewer for the attention. As the reviewer pointed out we have erased the duplicated sentence below. 

“The cannulation performance of each participant was video recorded.” to “ ” In line 171. 

4. Line 212. "was all comparable" should read "were all comparable"

Response: We have revised the sentence as the reviewer pointed out in the revised Result section. 

“Training year, baseline clinical experience of palpated radial artery cannulation, and ultrasound-guided central line cannulation were all comparable between the two groups (Table 1).” In lines 216-218. 

5. Supplement table 4. It is not clear how to read this table. Because these are all questions, maybe it would be better to create "yes" and "no" columns for the control and the simulation group and then fill in the number of responses

Response: As the reviewer suggested we have revised the supplement table 4. Please refer to the attached file < Supplement table 4>.

6. Line 284. The word "enhance" does not fit in the sentence well. Consider using enhancement, improvement, etc.

Response: We appreciate the reviewer for the comment. We have changed the description in the revised Discussion section. 

“Ten simulation sessions were sufficient to demonstrate improvement in skill proficiency, which is in agreement with the findings of previous study suggesting range of six to ten simulation training sessions to demonstrate competence in ultrasound-guided radial artery cannulation.” In lines 288-291.

7. Line 302. The 50% first pass success rate in novices is not higher than expected. Ueda et al in A randomized controlled trial of radial artery cannulation guided by doppler vs palpation vs ultrasound, achieved a first pass of 53% in anesthesia trainees with < 5 USG radial artery cannulations. levin et al in Use of ultrasound guidance in the insertion of radial artery catheters, Crit Care Med 2003, achieved a first pass success rate of 62% in operators with ultrasound guided central venous catheter experience but novices in radial artery cannulation with ultrasound guidance

Response: We appreciate the reviewer for the comment. We have changed the description in the revised Discussion section according to your comments.

“On the other hand, the residents in the control group showed a similar first attempt success rates of 53% in anesthesia trainees with less than five experiences in ultrasound-guided radial artery cannulation (1) and 62% in anesthetists with ultrasound-guided central vein insertion experience but novice to ultrasound-guided radial artery cannulation.(2) Among successful first attempt cases, the control group showed less success in terms of dynamic needle-tip positioning ability during the procedure, and this group achieved lower overall procedure level scores than the simulation group.” In lines 307-313.

Reference

1. Ueda K, Bayman E, Johnson C, Odum N, Lee JJ. A randomised controlled trial of radial artery cannulation guided by Doppler vs palpation vs ultrasound. Anaesthesia. 2015;70(9):1039-44.

2. Levin PD, Sheinin O, Gozal Y. Use of ultrasound guidance in the insertion of radial artery catheters. Critical care medicine. 2003;31(2):481-4.

We hope our additionally revised manuscript will better meet your requirements for publication. We thank the editor and the reviewer of Plos One once again for their constructive review of our paper. 

Warm regards,

Jeong Jin Min, M.D., Ph.D. 

Samsung Medical Center,

Sungkyunkwan University School of Medicine

---

## [Editor Report · Decision Letter 2]

29 May 2020

Simulation-based training using a vessel phantom effectively improved first attempt success and dynamic needle-tip positioning ability for ultrasound-guided radial artery cannulation in real patients: an assessor-blinded randomized controlled study

PONE-D-19-35381R2

Dear Dr. Jeong Jin Min

We are pleased to inform you that your manuscript has been judged scientifically suitable for publication and will be formally accepted for publication once it complies with all outstanding technical requirements.

With kind regards,

Jun Takeshita, M.D., Ph.D.

Guest Editor

PLOS ONE
---

## [Editor Report · Acceptance letter]

2 Jun 2020

PONE-D-19-35381R2 

Simulation-based training using a vessel phantom effectively improved first attempt success and dynamic needle-tip positioning ability for ultrasound-guided radial artery cannulation in real patients: an assessor-blinded randomized controlled study 

Dear Dr. Min:

I'm pleased to inform you that your manuscript has been deemed suitable for publication in PLOS ONE. Congratulations! Your manuscript is now with our production department. 

Kind regards, 

on behalf of

Dr. Jun Takeshita 

Guest Editor

PLOS ONE